# Arresting failure propagation in buildings through collapse isolation

Nirvan Makoond[1], Andri Setiawan[1], Manuel Buitrago[1] & Jose M. Adam[1 ✉]

Several catastrophic building collapses[1–5] occur because of the propagation of local-initial failures[6,7]. Current design methods attempt to completely prevent collapse after initial failures by improving connectivity between building components. These measures ensure that the loads supported by the failed components are redistributed to the rest of the structural system[8,9]. However, increased connectivity can contribute to collapsing elements pulling down parts of a building that would otherwise be unaffected[10]. This risk is particularly important when large initial failures occur, as tends to be the case in the most disastrous collapses[6]. Here we present an original design approach to arrest collapse propagation after major initial failures. When a collapse initiates, the approach ensures that specific elements fail before the failure of the most critical components for global stability. The structural system thus separates into different parts and isolates collapse when its propagation would otherwise be inevitable. The effectiveness of the approach is proved through unique experimental tests on a purposely built full-scale building. We also demonstrate that large initial failures would lead to total collapse of the test building if increased connectivity was implemented as recommended by present guidelines. Our proposed approach enables incorporating a last line of defence for more resilient buildings.

Disasters recorded from 2000 to 2019 are estimated to have caused economic losses of US$2.97 trillion and claimed approximately 1.23 million lives[11]. Most of these losses can be attributed to building collapses[12], which are often characterized by the propagation of local-initial failures[13] that can arise because of extreme or abnormal events such as earthquakes[13–16], floods[17–20], storms[21,22], landslides[23,24], explosions[25], vehicle impacts[26] and even construction or design errors[6,26]. As the world faces increasing trends in the frequency and intensity of extreme events[27,28], it is arguably now more important than ever to design robust structures that are insensitive to initial damage[13,29], irrespective of the underlying threat causing it.

Most robustness design approaches used at present[8,9,30,31] aim to completely prevent collapse initiation after a local failure by providing extensive connectivity within a structural system. Although these measures can ensure that the load supported by a failed component is redistributed to the rest of the structure, they are neither viable nor sustainable when considering larger initial failures[13,25,32]. In these situations, the implementation of these approaches can even result in collapsing parts of the building pulling down the rest of the structure[10]. The fact that several major collapses have occurred because of large initial failures[6] raises serious concerns about the inadequacy of the current robustness measures.

Traditionally, research in this area has focused on preventing collapse initiation after initial failures rather than on preventing collapse propagation. This trend dates back to the first impactful studies in the field of structural robustness, which were performed after a lack of connectivity enabled the progressive collapse of part of the Ronan Point tower in 1968 (ref. 33). Although completely preventing any collapse is

certainly preferable to limiting the extent of a collapse, the occurrence of unforeseeable incidents is inevitable[34] and major building collapses keep occurring[1–3].

Here we present an original approach for designing buildings to isolate the collapse triggered by a large initial failure. The approach, which is based on controlling the hierarchy of failures in a structural system, is inspired by how lizards shed their tails to escape predators[35]. The proposed hierarchy-based collapse isolation design ensures sufficient connectivity for operational conditions and after local-initial failures for which collapse initiation can be completely prevented through load redistribution. These local-initial failures can even be greater than those considered by building codes. Simultaneously, the structural system is also designed to separate into different parts and isolate a collapse when its propagation would otherwise be inevitable. As in the case of lizard tail autotomy[35], this is achieved by promoting controlled fracture along predefined segment borders to limit failure propagation. In this work, hierarchy-based collapse isolation is applied to framed building structures. Developing this approach required a precise characterization of the collapse propagation mechanisms that need to be controlled. This was achieved using computational simulations that were validated through a specifically designed partial collapse test of a full-scale building. The obtained results demonstrate the viability of incorporating hierarchy-based collapse isolation in building design.

## Hierarchy-based collapse isolation

Hierarchy-based collapse isolation design makes an important distinction between two types of initial failures. The first, referred to as small

[1]ICITECH, Universitat Politècnica de València, Valencia, Spain. ✉e-mail: joadmar@upv.es

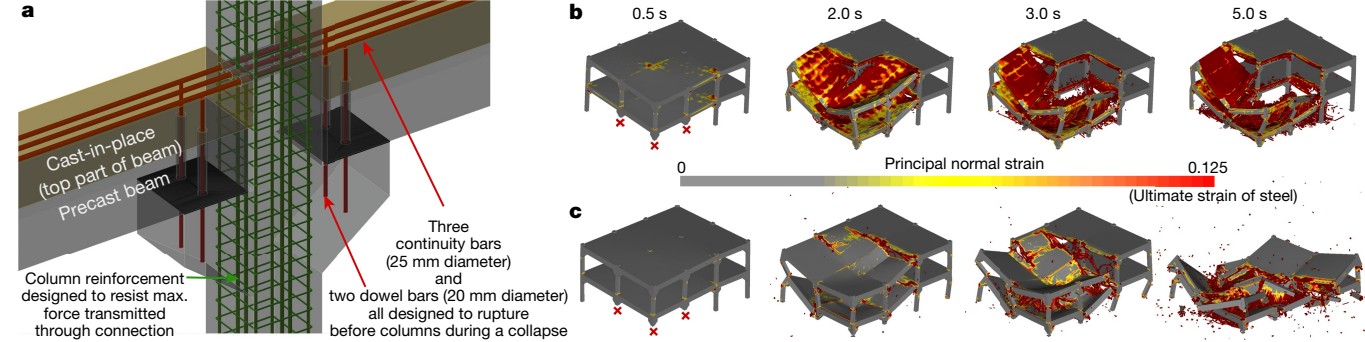

**Fig. 1 | Hierarchy-based collapse isolation and computational simulations.**
**a**, Partial-strength beam–column connection optimized for hierarchy-based collapse isolation. **b**, Partial collapse of a building designed for hierarchy-based collapse isolation (design H) after the loss of a corner column and two penultimate-edge columns. **c**, Total collapse of conventional building design (design C) after the same large initial failure scenario.

initial failures, includes all failures for which it is feasible to completely prevent the initiation of collapse by redistributing loads to the remaining structural system. The second type of initial failure, referred to as large initial failures, includes more severe failures that inevitably trigger at least a partial collapse.

The proposed design approach aims to (1) arrest unimpeded collapse propagation caused by large initial failures and (2) ensure the ability of a building to develop alternative load paths (ALPs) to prevent collapse initiation after small initial failures. This is achieved by prioritizing a specific hierarchy of failures among the components on the boundary of a moving collapse front.

Buildings are complex three-dimensional structural systems consisting of different components with very specific functions for transferring loads to the ground. Among these, vertical load-bearing components such as columns are the most important for ensuring global structural stability and integrity. Therefore, hierarchy-based collapse isolation design prevents the successive failure of columns, which would otherwise lead to catastrophic collapse. Although the exact magnitude of dynamic forces transmitted to columns during a collapse process is difficult to predict, these forces are eventually limited by the connections between columns and floor systems. In the proposed approach, partial-strength connections are designed to limit the magnitude of transmitted forces to values that are lower than the capacity of columns to resist unbalanced forces (see section 'Building design'). This requirement guarantees a specific hierarchy of failures during collapse, whereby connection failures always occur before column failures. As a result, the collapse following a large initial failure is always restricted to components immediately adjacent to those directly involved in the initial failure. However, it is still necessary to ensure a lower bound on connection strengths to activate ALPs after small initial failures. Therefore, cost-effective implementation of hierarchy-based collapse isolation design requires finding an optimal balance between reducing the strength of connections and increasing the capacity of columns.

To test and verify the application of our proposed approach, we designed a real 15 m × 12 m precast reinforced concrete building with two 2.6-m-high floors. This basic geometry represents a building size that can be built and tested at full-scale while still being representative of current practices in the construction sector. The structural type was selected because of the increasing use of prefabricated construction for erecting high-occupancy buildings such as hospitals and malls because of several advantages in terms of quality, efficiency and sustainability[36].

The collapse behaviour of possible design options (Extended Data Fig. 1) subjected to both small and large initial failures was investigated using high-fidelity collapse simulations (Fig. 1) based on the applied element method (AEM; see section 'Modelling strategy'). The ability of these simulations to accurately represent collapse phenomena for the type of building being studied was later validated by comparing its predictions to the structural response observed during a purposely designed collapse test of a full-scale building (Extended Data Fig. 2 and Supplementary Video 7).

Following the preliminary design of a structure to resist loads suitable for office buildings, two building design options considering different robustness criteria were further investigated (see section 'Building design'). The first option, design H (hierarchy-based), uses optimized partial-strength connections and enhanced columns (Fig. 1a) to fulfil the requirements of hierarchy-based collapse isolation design. The second option, design C (conventional), is strictly based on code requirements and provides a benchmark comparison for evaluating the effectiveness of the proposed approach. It uses full-strength connections to improve robustness as recommended in current guidelines[37] and building codes[8,9].

Simulations predicted that both design H and design C could develop stable ALPs that are able to completely prevent the initiation of collapse after small initial failure scenarios that are more severe than those considered in building codes[8,9] (Extended Data Fig. 3).

When subjected to a larger initial failure, simulations predict that design H can isolate the collapse to only the region directly affected by the initial failure (Fig. 1b). By contrast, design C, with increased connectivity, causes collapsing elements to pull down the rest of the structure, leading to total collapse (Fig. 1c). These two distinct outcomes demonstrate that the prevention of unimpeded collapse propagation can only be ensured when hierarchy-based collapse isolation is implemented (Extended Data Fig. 4 and Supplementary Video 1).

## Testing a full-scale precast building

To confirm the expected performance improvement that can be achieved with the hierarchy-based collapse isolation design, a full-scale building specimen corresponding to design H was purposely built and subjected to two phases of testing as part of this work (Fig. 2a and Supplementary Information Sections 1 and 2). The precast structure was constructed with continuous columns cast together with corbels (Supplementary Video 4). The columns were cast with prepared dowel bars and sleeves for placing continuous top beam reinforcement bars through columns (Fig. 2b,c). The bars used for these two types of reinforcing element (Fig. 1a) were specifically selected to produce partial-strength connections. These connections are strong enough for the development of ALPs after small initial failures but weak enough to enable hierarchy-based collapse isolation after large initial failures.

After investigating different column-removal scenarios from different regions of the test building (see section 'Experiment and monitoring design', Extended Data Fig. 5 and Supplementary Video 2), two phases of testing were defined to capture relevant collapse-related phenomena

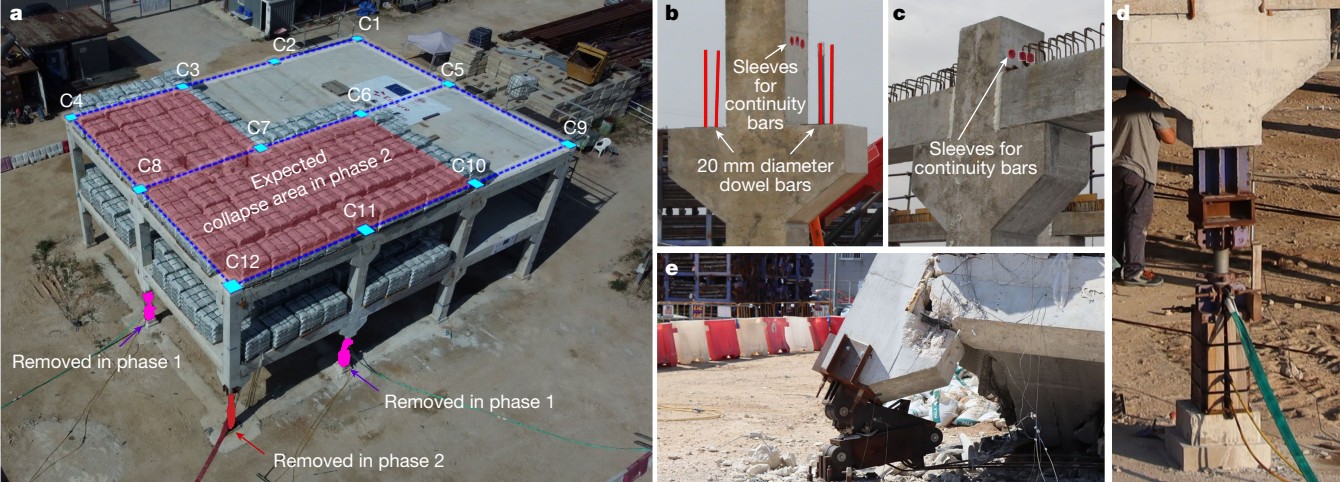

**Fig. 2 | Building design and testing. a**, Full-scale precast concrete structure and columns removed in different testing phases. The label used for each column is shown. The location of beams connecting the different columns is indicated by the dotted lines above the second-floor level. The expected collapse area in the second phase of testing is indicated. **b**, Typical first-floor connection before placement of beams during construction. **c**, Typical second-floor connection after placement of precast beams during construction. Both **b** and **c** show columns with two straight precast beams on either side (C2, C3, C6, C7, C10 and C11). **d**, Device used for quasi-static removal of two columns in the first phase of testing. **e**, Three-hinged mechanism used for dynamic removal of corner column in the second phase of testing.

and validate the effectiveness of hierarchy-based collapse isolation. Separating the test into two phases allowed two different aspects to be analysed: (1) the prevention of collapse initiation after small initial failures and (2) the isolation of collapse after large initial failures.

Phase 1 involved the quasi-static removal of two penultimate-edge columns using specifically designed removable supports (Fig. 2d and Extended Data Fig. 6). This testing phase corresponds to a small initial failure scenario for which design H was able to develop ALPs to prevent collapse initiation. Phase 2 reproduced a large initial failure through the dynamic removal of the corner column found between the two previously removed columns using a three-hinged collapsible column (Fig. 2e).

During both testing phases, a distributed load (11.8 kN m$^{-2}$) corresponding to almost twice the magnitude specified in Eurocodes[38] for accidental design situations (6 kN m$^{-2}$) was imposed on bays expected to collapse in phase 2 (Fig. 2a and Supplementary Video 5). Predictive simulations indicated that the failure mode and overall collapse would be almost identical when comparing this partial loading configuration with that in which the entire building is loaded (Supplementary Video 3). However, the partial loading configuration turns out to be more demanding for the part of the structure expected to remain upright as evidenced by the greater drifts it produces during collapse (see section 'Experiment and monitoring design' and Extended Data Fig. 7). The structural response during all phases of testing was extensively monitored with an array of different sensors (see section 'Experiment and monitoring design' and Supplementary Information Section 3) that provided the information used as a basis for the analyses presented in the following sections.

## Preventing collapse initiation

Collapse initiation was completely prevented after the removal of two penultimate-edge columns in phase 1 of testing (Fig. 3a), demonstrating that design H complies with the robustness requirements included in current building standards[8,9,39]. As this initial failure scenario is more severe than those considered by standardized design methods[8,9,30], it represents an extreme case for which ALPs are still effective. As such, the outcome of phase 1 demonstrates that implementing hierarchy-based collapse isolation design does not impair the ability of this structure to prevent collapse initiation.

Analysis of the structural response during phase 1 (Supplementary Information Section 4) shows that collapse was prevented because of the redistribution of loads through the beams (Fig. 3b,c), columns (Extended Data Fig. 8) and slabs (Supplementary Report 4) adjacent to the removed columns. The beams bridging over the removed columns sustained loads through flexural action, as evidenced by the magnitude of the vertical displacement recorded at the removal locations (Fig. 3b). These values were far too small to allow the development of catenary forces, which only begin to appear when displacements exceed the depth of the beam[40].

The flexural response of the structure after the loss of two penultimate-edge columns was only able to develop because of the specific reinforcement detailing introduced in the design. This was verified by the increase in tensile strains recorded in the continuous beam reinforcement close to the removed column (Fig. 3c) and in ties placed between the precast hollow-core planks in the floor system close to column C7 (Supplementary Information Section 4). The latter also proves that the slabs contributed notably to load redistribution after column removal.

In general, the structure experienced only small movements and suffered very little permanent damage during phase 1 (Supplementary Information Section 4), despite the high imposed loads used for testing. The only reinforcement bars showing some signs of yielding were the continuous reinforcement bars of beams close to the removed columns (Fig. 3c).

## Arresting collapse propagation

Following the removal of two penultimate-edge columns in phase 1, the sudden removal of the C12 corner column in phase 2 triggered a collapse that was arrested along the border delineated by columns C3, C7, C6 and C10 (Fig. 4a–d and Supplementary Video 6). Thus, the viability of hierarchy-based collapse isolation design is confirmed.

During the initial stages following the removal of C12, the collapsing bays next to this column pulled up the columns on the opposite corner of the building (columns C1, C3 and C6). During this process, column C7 behaves like a pivot point, experiencing a significant increase in compressive forces (Fig. 4e and Supplementary Information Section 5). This phenomenon was enabled by the connectivity between collapsing

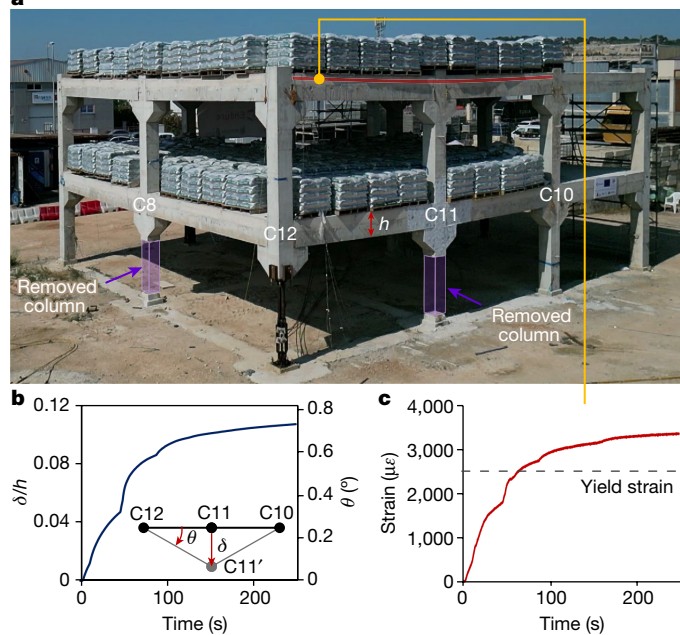

**Fig. 3 | When ALPs are effective. a**, Test building during phase 1 of testing after removal of columns C8 and C11. The beam depth (*h*) used to compute the ratio plotted in **b** is shown and the location of the strain measurement plotted in **c** is indicated. **b**, Evolution of beam deflection expressed as a ratio of beam depth at the location of removed column C11. The chord rotation of the beams bridging over this removed column is also indicated using a secondary vertical axis. **c**, Strain increase in continuity reinforcement in the second-floor beam between C12 and C11.

parts and the rest of the structure. If allowed to continue, this could have led to successive column failures and unimpeded collapse propagation. However, during the test, the rupture of continuous reinforcement bars (Fig. 4c) occurred as the connections failed and halted the transmission of forces to columns. These connection failures occurred before any column failures, as intended by the hierarchy-based collapse isolation design of the structural system. Specifically, this type of connection failure occurred at the junctions with the two columns (C7 and C10) immediately adjacent to the failure origin (around C8, C11 and C12), effectively segmenting the structure along the border shown in Fig. 4b. Segmentation along this border was completed by the total separation of the floor system, which was enabled by the debonding of slab reinforcements at the segment border (Fig. 4d and Supplementary Video 8).

Observing the building drift measured at the top of column C9 (Fig. 4f) enabled us to better understand the nature of forces acting on the building further away from the collapsing region. The initial motion shows the direction of pulling forces generated by the collapsing elements (Fig. 4g). This drift peaks very shortly after the point in time when separation of the collapsing parts occurs (Fig. 4f). After this peak, the upright part of the structure recoiled backwards and experienced an attenuated oscillatory motion before finding a new stable equilibrium (Fig. 4g). The magnitude of the measured peak drift is comparable to the drift limits considered in seismic regions when designing against earthquakes with a 2,500-year return period[41] (Supplementary Information Section 5). This indicates that the upright part of the structure was subjected to strong dynamic horizontal forces as it was effectively tugged by the collapsing elements falling to the ground. The building would have failed because of these unbalanced forces had hierarchy-based collapse isolation design not been implemented.

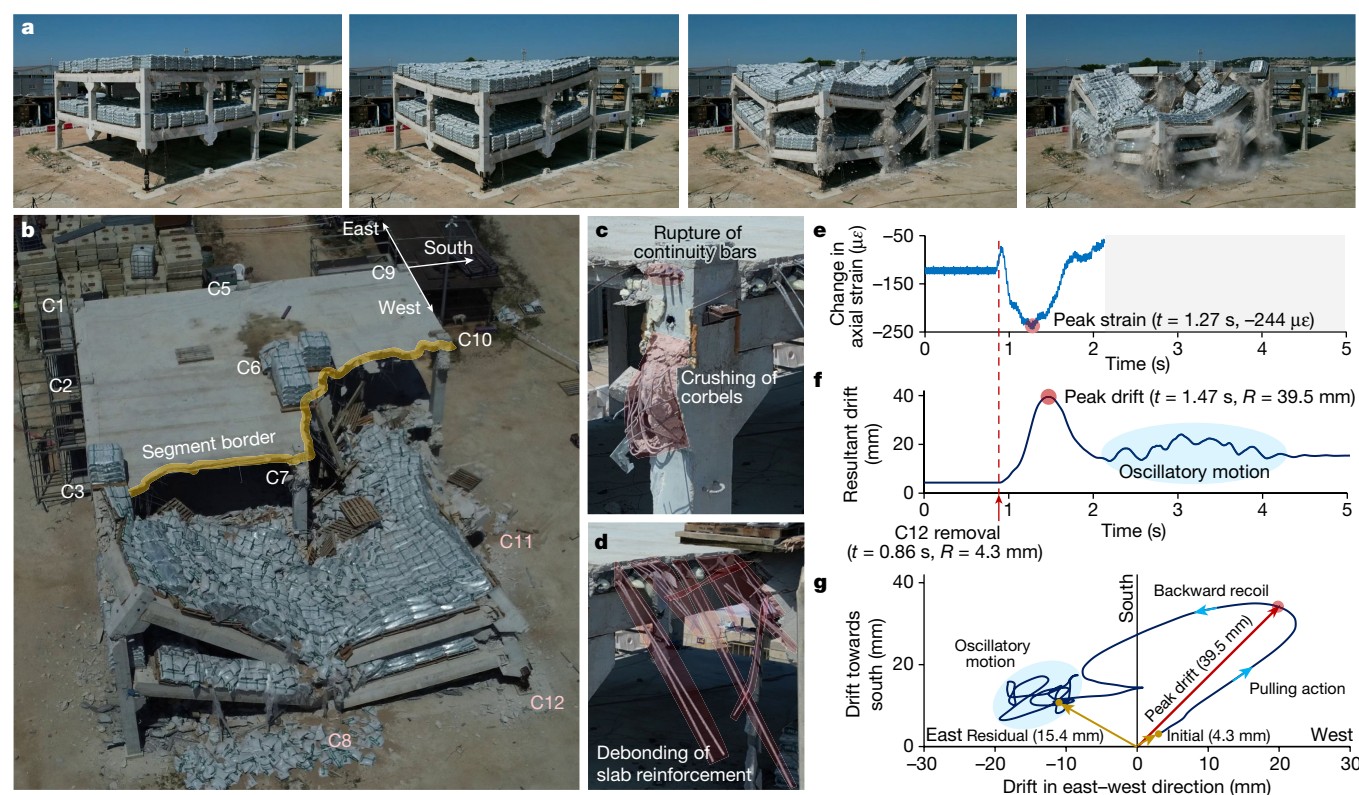

**Fig. 4 | Segmentation and partial collapse. a**, Collapse sequence during phase 2 of testing. **b**, Partial collapse of full-scale test building (design H) after the removal of three columns. The segment border in which collapse propagation was arrested is indicated. The axes shown at column C9 correspond to those used in **f** to indicate the changing direction of the resultant drift measured at this location. **c**, Failure of beam–column connections at collapse border. **d**, Debonding of reinforcement in the floor at collapse border. **e**, Change in average axial strains measured in column C7. A negative change represents an increase in compressive strains. **f**, Magnitude of resultant drift measured at C9. **g**, Change in direction of resultant drift measured at C9. The initial drift after phase 1 of testing and the residual drift after the upright part of the building stabilized are also shown in the plot.

The upright building segment suffered permanent damages as evidenced by the residual drift recorded at the top of column C9 (Fig. 4g). This is further corroborated by the fact that several reinforcement bars in this part of the structure yielded, particularly in areas close to the segment border (Supplementary Report 5). Despite the observed level of damage, safe evacuation and rescue of people from this building segment would still be possible after an extreme event, saving lives that would have been lost had a more conventional robustness design (design C) been used instead.

## Discussion and future outlook

Our results demonstrate that the extensive connectivity adopted in conventional robustness design can lead to catastrophic collapse after large initial failures. To address this risk, we have developed and tested a collapse isolation design approach based on controlling the hierarchy of failures occurring during the collapse. Specifically, it is ensured that connection failures occur before column failures, mitigating the risk of collapse propagation throughout the rest of the structural system. The proposed approach has been validated through the partial collapse test of a full-scale precast building, showing that propagating collapses can be arrested at low cost without impairing the ability of the structure to completely prevent collapse initiation after small initial failures.

The reported findings show a last line of defence against major building collapses due to extreme events. This paves the way for the proposed solution to be developed, tested and implemented in different building types with different building elements. This discovery opens opportunities for robustness design that will lead to a new generation of solutions for avoiding catastrophic building collapses.

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

# Methods

## Building design

Our hierarchy-based collapse isolation approach ensures buildings have sufficient connectivity for operational conditions and small initial failures, yet separate into different parts and isolate a collapse after large initial failures. We chose a precast construction as our main structural system for our case study. A notable particularity of precast systems compared with cast-in-place buildings is that the required construction details can be implemented more precisely. We designed and systematically investigated two precast building designs: designs H and C.

**Design H.** Design H is our building design in which the hierarchy-based collapse isolation approach is applied. Design H was achieved after several preliminary iterations by evaluating various connections and construction details commonly adopted in precast structures. The final design comprises precast columns with corbels connected to a floor system (partially precast beams and hollow-core slabs) through partial-strength beam–column connections (Extended Data Fig. 1 and Supplementary Information Section 1). This partial-strength connection was achieved by (1) connecting the bottom part of the beam (precast) to optimally designed dowel bars anchored to the column corbels and (2) passing continuous top beam bars through the columns. With this partial-strength connection, we have more direct control over the magnitude of forces being transferred from the floor system to the columns, which is a key aspect for achieving hierarchy-based collapse isolation. The hierarchy of failures was initially implemented through the beam–column connections (local level) and later verified at the system (global) level.

At the local level, three main components are designed according to the hierarchy-based concept: (1) top continuity bars of the beams; (2) dowel bars connecting beams to corbels; and (3) columns.

1.  Top continuity bars of beams: To allow the structural system to redistribute the loads after small initial failures, top reinforcement bars in all beams were specifically designed to fulfil structural robustness requirements (Extended Data Fig. 3). Particularly, we adopted the prescriptive tying rules (referred to as Tie Forces) of UFC 4-023-03 (ref. 9) to perform the design of the ties. The required tie strength $F_i$ in both the longitudinal and transverse directions for the internal beams is expressed as

    $$F_i = 3w_{\text{F}}L_1$$

    For the peripheral beams, the required tie strength $F_P$ is expressed as

    $$F_P = 6w_{\text{F}}L_1L_P + 3W_{\text{C}}$$

    with

    $$w_{\text{F}} = 1.2D + 0.5L$$

    where $w_{\text{F}}$ = floor load (in kN m$^{-2}$); $D$ = dead load (in kN m$^{-2}$); $L$ = live load (in kN m$^{-2}$); $L_1$ = greater of the distances between the centres of the columns, frames or walls supporting any two adjacent floor spaces in the direction under consideration (in m); $L_P$ = 1.0 m; and $W_{\text{C}}$ = 1.2 times dead load of cladding (neglected in this design).

    These required tie strengths are fulfilled with three bars (20 mm diameter) for the peripheral beams and three bars (25 mm diameter) for the internal beams. These required reinforcement dimensions were implemented through the top bars of the beam and installed continuously (lap-spliced, internally, and anchored with couplers at the ends) throughout the building (Extended Data Fig. 1).

2.  Dowel bars connecting the beam and corbel of the column: The design of the dowel bars is one of the key aspects in achieving

partial-strength connections that fail at a specific threshold to enable segmentation. These dowel bars would control the magnitude of the internal forces between the floor system and column while allowing for some degree of rotational movement. The dowels were designed to resist possible failure modes using expressions proposed in the fib guidelines[37]. Several possible failure modes were checked: splitting of concrete around the dowel bars, shear failure of the dowel bars and forming a plastic hinge in the dowel. The shear capacity of a dowel bar loaded in pure shear can be determined according to the Von Mises yield criterion:

$$F_{vR\_s} = \frac{1}{\sqrt{3}} f_{\text{yd}} A_{\text{s}}$$

where $f_{\text{yd}}$ is the design yield strength of the dowel bar and $A_{\text{s}}$ is the cross-sectional area of the dowel bar. In case of concrete splitting failure, the highly concentrated reaction transferred from the dowel bar shall be designed to be safely spread to the surrounding concrete. The strut and tie method is recommended to perform such a design[42]. If shear failure and splitting of concrete do not occur prematurely, the dowel bar will normally yield in bending, indicated by the formation of a plastic hinge. This failure mode is associated with a significant tensile strain at the plastic hinge location of the dowel bar and the crushing of concrete around the compression part of the dowel. The shear resistance achieved at this state for dowel (ribbed) bars across a joint of a certain width (that is, the neoprene bearing) can be expressed as

$$F_{vR\_y} = \alpha_0 \alpha_e \Phi^2 \sqrt{f_{\text{cd,max}} f_{\text{yd,red}}} + \mu \sigma_{\text{sn}} A_{\text{s}}$$

with

$$\alpha_e = \sqrt{1 + (\varepsilon \alpha_0)^2} - \varepsilon \alpha_0$$

$$\varepsilon = 3\frac{e}{\Phi} \sqrt{\frac{f_{\text{cd,max}}}{f_{\text{yd,red}}}}$$

where $\alpha_0$ is a coefficient that considers the bearing strength of concrete and can be taken as 1.0 for design purposes, $\alpha_e$ is a coefficient that considers the eccentricity, $e$ is the load eccentricity and shall be computed as the half of the joint width (half of the neoprene bearing thickness), $\Phi$ and $A_{\text{s}}$ are the diameter and the cross-sectional area of the dowel bar, respectively, $f_{\text{cd,max}}$ is the design concrete compressive strength at the stronger side, $\sigma_{\text{sn}}$ is the local axial stress of the dowel bar at the interface location, $f_{\text{yd,red}} = f_{\text{yd}} - \sigma_{\text{sn}}$ is the design yield strength available for dowel action, $f_{\text{yd}}$ is the yield strength of the dowel bar and $\mu$ is the coefficient of friction between the concrete and neoprene bearing. By performing the checks on these three possible failure modes, we selected the final (optimum) design with a two dowel bars (20 mm diameter) configuration.

3.  Columns: The proposed hierarchy-based approach requires columns to have adequate capacity to resist the internal forces transmitted by the floor system during a collapse. By fulfilling this strength hierarchy, we can ensure and control that failure happens at the connections first before the columns fail, thus preventing collapse propagation. The columns were initially designed according to the general procedure prescribed by building standards. Then, the resulting capacity was verified using the modified compression field theory (MCFT)[43] to ensure that it was higher than the maximum expected forces transmitted by the connection to the floor system. MCFT was derived to consistently fulfil three main aspects: equilibrium of forces, compatibility and rational stress–strain relationships of cracked concrete expressed as average stresses and strains. The principal compressive stress in the concrete $f_{c2}$ is expressed not only

as a function of the principal compressive strain $\varepsilon_2$ but also of the co-existing principal tensile strain $\varepsilon_1$, known as the compression softening effect:

$$f_{c2} = f_{c2\max}\left[2\left(\frac{\varepsilon_2}{\varepsilon_{c'}}\right) - \left(\frac{\varepsilon_2}{\varepsilon_{c'}}\right)^2\right]$$

with

$$\frac{f_{c2\max}}{f_c'} = \frac{1}{0.8 - 0.34\frac{\varepsilon_1}{\varepsilon_{c'}}} \le 1.0$$

where $f_{c2\max}$ is the peak concrete compressive strength considering the perpendicular tensile strain, $f_c'$ is the uniaxial compressive strength, and $\varepsilon_{c'}$ is the peak uniaxial concrete compressive strain and can be taken as −0.002. In tension, concrete is assumed to behave linearly until the tensile strength is achieved, followed by a specific decaying function[43]. Regarding aggregate interlock, the shear stress that can be transmitted across cracks $v_{ci}$ is expressed as a function of the crack width $w$, and the required compressive stress on the crack $f_{ci}$ (ref. 44):

$$v_{ci} = 0.18v_{ci\max} + 1.64f_c' - 0.82\frac{f_{ci}^2}{v_{ci\max}}$$

with

$$v_{ci\max} = \frac{\sqrt{f_c'}}{0.31 + \frac{24w}{(a+16)}}$$

where $a$ refers to the maximum aggregate size in mm and the stresses are expressed in MPa. The MCFT analytical model was implemented to solve the sectional and full-member response of beams and columns subjected to axial, bending and shear in Response 2000 software (open access)[45,46]. In Response 2000, we input key information, including the geometries of the columns, reinforcement configuration and the material definition for the concrete and the reinforcing bars. Based on this information, we computed the $M–V$ (moment and shear interaction envelope) and $M–N$ (moment and axial interaction envelope) diagrams that represent the capacity of the columns. The results shown in Extended Data Fig. 4 about the verification of the demand and capacity envelopes were obtained using the analytical procedure described here.

At the global level, the initially collapsing regions of the building generate a significant magnitude of dynamic unbalanced forces. The rest of the building system must collectively resist these unbalanced forces to achieve a new equilibrium state. Depending on the design of the structure, this phenomenon can lead to two possible scenarios: (1) major collapse due to failure propagation or (2) partial collapse only of the initially affected regions. The complex interaction between the three-dimensional structural system and its components must be accounted for to evaluate the structural response during collapse accurately. Advanced computational simulations, described in the 'Modelling strategy' section, were adopted to analyse the global building to verify that major collapse can be prevented. The final design obtained from the local-level analysis (top continuity bars, dowel bars and columns) was used as an input for performing the global computational simulations. Certain large initial failures deemed suitable for evaluating the performance of this building were simulated. In case failure propagation occurs, the original hierarchy-based design must be further adapted. An iterative process is typically required involving several simulations with various building designs to achieve an optimum result that balances the cost and desired collapse performance. The final iteration of design H, which fulfils both the local and global hierarchy checks, is provided in Extended Data Fig. 1.

**Design C.** Design C is a conventional building design that complies with current robustness standards but does not explicitly fulfil our hierarchy-based approach. The same continuity bars used in design H were used in design C. We adopted a full-strength connection as recommended by the fib guideline[37]. The guideline promotes full connectivity to enhance the development of alternative load paths for preventing collapse initiation. In design C, we used a two dowel bars (32 mm diameter) configuration to ensure full connectivity when the beams are working at their maximum flexural capacity. Another main difference was that the columns in design C were designed according to codes and current practice (optimal solution) without explicitly checking that hierarchy-based collapse isolation criteria are fulfilled. The final design of the columns and connections adopted in design C is provided in Extended Data Fig. 1.

## Modelling strategy

We used the AEM implemented in the Extreme Loading for Structures software to perform all the computational simulations presented in this study[47] (Extended Data Figs. 2–5 and 7 and Supplementary Videos 1, 2, 3 and 7). We chose the AEM for its ability to represent all phases of a structural collapse efficiently and accurately, including element separation (fracture), contact and collision[47]. The method discretizes a continuum into small, finite-size elements (rigid bodies) connected using multiple normal and shear springs distributed across each element face. Each element has six degrees of freedom, three translational and three rotational, at its centre, whereas the behaviour of the springs represents all material constitutive models, contact and collision response. Despite the simplifying assumptions in its formulation[48], its ability to accurately account for large displacements[49], cyclic loading[50], as well as the effects of element separation, contact and collision[51] has been demonstrated through many comparisons with experimental and theoretical results[47].

**Geometric and physical representations.** We modelled each of the main structural components of the building separately, including the columns, beams, corbels and hollow-core slabs. We adopted a consistent mesh size with an average (representative) size of 150 mm. Adopting this mesh configuration resulted in a total number of 98,611 elements. We defined a specialized interface with no tensile or shear strength between the precast and cast-in-situ parts to allow for localized deformations that occur at these locations. The behaviour of the interface was mainly governed by a friction coefficient of 0.6, which was defined according to concrete design guidelines[52–54]. The normal stiffness of these interfaces corresponded to the stiffness of the concrete cast-in-situ topping. The elastomeric bearing pads supporting the precast beams on top of the corbels were also modelled with a similar interface having a coefficient of friction of 0.5 (ref. 55).

**Element type and constitutive models.** We adopted an eight-node hexahedron (cube) element with the so-called matrix-springs connecting adjacent cubes to model the concrete parts. We adopted the compression model in refs. 56,57 to simulate the behaviour of concrete under compression. Three specific parameters are required to define the response envelope: the initial elastic modulus, the fracture parameter and the compressive plastic strain. For the behaviour in tension, the spring stiffness is assumed to be linear (with the initial elastic modulus) until reaching the cracking point. The shear behaviour is considered to remain linear up to the cracking of the concrete. The interaction between normal compressive and shear stress follows the Mohr–Coulomb failure criterion. After reaching the peak, the shear stress is assumed to drop to a certain residual value affected by the aggregate interlock and friction at the cracked surface. By contrast,

under tension, both normal and shear stresses drop to zero after the cracking point. The steel reinforcement bars were simulated as a discrete spring element with three force components: the normal spring takes the principal/normal forces parallel to the rebar, and two other springs represent the reinforcement bar in shear (dowelling). Three distinct stages are considered: elastic, yield plateau and strain hardening. A perfect bond behaviour between the concrete and the reinforcement bars was adopted. We assigned the material properties based on the results of the laboratory tests performed on reinforcement bars and concrete cylinders (Supplementary Information Section 2).

**Boundary conditions and loading protocol.** We assumed that all the ground floor columns are fully restrained in all six degrees of freedom at the base location. This assumption is reasonable, as we expected that the footing would provide sufficient rigidity to constrain any significant deformations. We assigned the reflecting domain boundaries to allow a realistic representation of the collapsing elements (debris) that might fall and rebound after hitting the ground. The ground level was assumed to be at the same elevation at which the column bases are restrained. We applied the additional imposed uniform distributed load as an extra volume of mass assigned to the slabs. To perform the column removal, we used the element removal feature that allows some specific designated elements to be immediately removed at the beginning of the loading stage. This represents a dynamic (sudden) removal, as we expected from the actual test.

Extended Data Tables 1 and 2 summarize all key parameters and assumptions adopted in the modelling process. To validate these assumptions for simulating the precast building designs described previously, it was ensured that the full-scale test performed as part of this work captured all relevant phenomena influencing collapse (large displacements, fracture, contact and collision).

### Experiment and monitoring design

We used computational simulations of design H subjected to different initial failure scenarios to define a suitable testing sequence and protocol. The geometry, reinforcement configurations, connection system and construction details of the purpose-built specimen representing design H are provided in Supplementary Information Section 1 and Supplementary Video 4.

**Initial failure scenarios.** Initial failure scenarios occurring in edge and corner regions of the building were prioritized for this study because they are usually exposed to a wider range of external threats[58–61]. After performing a systematic sensitivity study, we identified three critical scenarios (Extended Data Fig. 5 and Supplementary Video 2):
1. Scenario 1: a scenario involving a two-column failure—a corner column and the adjacent edge column. We determined that the required gravity loads to induce collapse equal 11.5 kN m$^{-2}$ and that partial collapse would occur locally.
2. Scenario 2: a scenario involving a three-column failure—two corner columns and the edge column in between the two corner columns. We determined that the required gravity loads to induce collapse equal 8.5 kN m$^{-2}$ and that segmentation (partially collapsing two bays) would take place only across one principal axis of the building.
3. Scenario 3: a scenario involving a three-column failure: one corner column and two edge columns on both sides of the corner column. We determined that the required gravity loads to induce collapse equal 7.0 kN m$^{-2}$ and that segmentation (partially collapsing three bays) would take place across both principal axes of the building.

Scenario 3 was ultimately chosen after considering three main aspects: (1) it requires the lowest gravity loads to trigger partial collapse; (2) the failure mode involves activating segmentation mechanisms in two principal axes of the building (more realistic collapse pattern); and (3) the ratio of the area of the intact part and the collapsed part was predicted to be 50:50, leading to the largest collapse area among the three scenarios.

**Testing phases.** To allow us to investigate the behaviour of the building specimen under small and large initial failures in only one building specimen, we decided to perform two separate testing phases. Phase 1 involved the quasi-static (gradual) removal of two edge columns (C8 and C11), whereas phase 2 involved the sudden removal of the corner column (C12) found between the columns removed in phase 1. A uniformly distributed load of 11.8 kN m$^{-2}$ was applied only on the bays directly adjacent to these three columns without loading the remaining bays (Supplementary Video 5). This was achieved by placing more than 8,000 sandbags in the designated bays on the two floors (the first- and second-floor slabs). We performed additional computational simulations to compare this partial loading configuration and loading of the entire building. The simulations indicated that both would have resulted in almost identical final collapse states (Extended Data Fig. 7 and Supplementary Video 3). However, the partial loading configuration introduced a higher magnitude of unbalanced moment to surrounding columns, which induces more demanding bending and shear in columns. Simulations confirmed that the lateral drift of the remaining part of the building would be higher when only three bays are loaded, indicating that its stability would be tested to a greater extent with this loading configuration (Extended Data Fig. 7).

**Specially designed elements to trigger initial failures.** We designed special devices to perform the column removal (Extended Data Fig. 6). For phase 1, we constructed two hanging concrete columns (C8 and C11) supported only on a vertical hydraulic jack. The pressure in the jack could be gradually released from a safe distance to remove the vertical reaction supporting the column. In phase 2, a three-steel-hinged column was used as the corner column. The middle part of the column represents a central hinge that was able to rotate if unlocked. During the second testing phase, we unlocked the hinge by pulling the column from outside the building using a forklift to induce a slight destabilization. This resulted in a sudden removal of the corner column C12 and the initiation of the collapse.

**Monitoring plan.** To monitor the structural behaviour, we heavily instrumented the building specimen with multiple sensors. A total of 57 embedded strain gauges, 17 displacement transducers and 5 accelerometers were placed at key locations in different parts of the structure (Extended Data Fig. 8 and Supplementary Information Section 3) during all phases of testing. The data from these sensors (Supplementary Information Sections 4 and 5) were complemented by the pictures and videos of the structural response captured by five high-resolution cameras and two drones (Supplementary Videos 6 and 8).

## Data availability

All experimental data recorded during testing of the full-scale building are available from Zenodo (https://doi.org/10.5281/zenodo.10698030)[62]. Source data are provided with this paper.

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

**Acknowledgements** This article is part of a project (Endure) that has received funding from the European Research Council (ERC) under the Horizon 2020 research and innovation programme of the European Union (grant agreement no. 101000396). We acknowledge the assistance of the following colleagues from the ICITECH-UPV institute in preparing and executing the full-scale building tests: J. J. Moragues, P. Calderón, D. Tasquer, G. Caredda, D. Cetina, M. L. Gerbaudo, L. Marín, M. Oliver and G. Sempértegui. We are also grateful to the Levantina, Ingeniería y Construcción S.L. (LIC) company for providing human resources and access to their facilities for testing. Finally, we thank A. Elfouly and Applied Science International for their support in performing simulations.

**Author contributions** N.M. prepared the main text, performed the computational simulations and validated the test results. A.S. analysed the experimental data, performed data curation and prepared the Methods section. M.B. contributed to the design of the building specimen, the design of the test and data curation. J.M.A. contributed to the design of the research methodology, supervised the research and was responsible for funding acquisition. N.M., A.S. and M.B. contributed to the execution of the experimental test and to preparing figures, extended data and supplementary information. All authors interpreted the test and simulation results and edited the paper.

**Competing interests** The authors declare no competing interests.

**Additional information**
**Correspondence and requests for materials** should be addressed to Jose M. Adam.

## General building layout

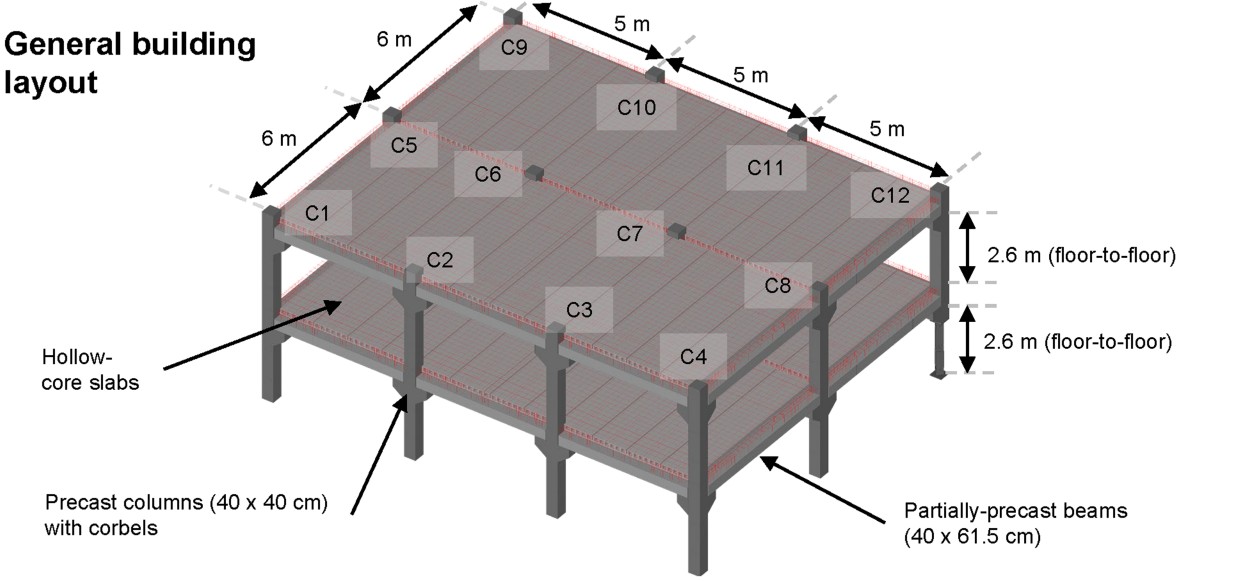

6 m
6 m

5 m
5 m
5 m

C9
C10
C5
C6
C11
C1
C12
C7
C2
C8
C3
C4

2.6 m (floor-to-floor)
2.6 m (floor-to-floor)

Hollow-core slabs

Precast columns (40 x 40 cm) with corbels

Partially-precast beams (40 x 61.5 cm)

---

## Design H ("Hierarchy-based")

Continuity bars:
3-Φ25 for internal beams
3-Φ20 for perimeter beams

Dowel bars:
2-Φ20

**Partial-strength connection**

**Column reinforcement configuration (example: C7)**

1st Floor — 8-Φ16

Stirrups:
0 – 0.8 m (height): Φ8-60 mm
0.8 – 1.8 m: Φ8-150 mm
1.8 – 3.2 m: Φ8-100 mm

Ground Floor — 8-Φ25

Stirrups:
0 – 0.8 m (height): Φ8-60 mm
0.8 – 1.8 m: Φ8-200 mm
1.8 – 3.2 m: Φ8-100 mm

Detail drawings showing the reinforcement configurations for other columns are provided in the Supplementary Information, Test Report (Appendix)

## Design C ("Conventional")

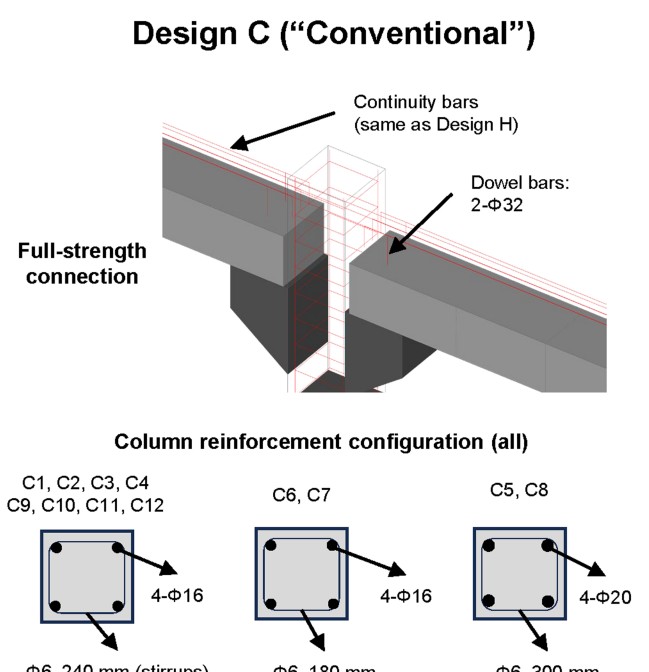

Continuity bars (same as Design H)

Dowel bars:
2-Φ32

**Full-strength connection**

**Column reinforcement configuration (all)**

C1, C2, C3, C4
C9, C10, C11, C12 — 4-Φ16

Φ6–240 mm (stirrups)

C6, C7 — 4-Φ16

Φ6–180 mm

C5, C8 — 4-Φ20

Φ6–300 mm

---

**Extended Data Fig. 1 | Summary of building designs.** General building layout, connection details, and reinforcement configurations of Design H ("Hierarchy-based") and Design C ("Conventional").

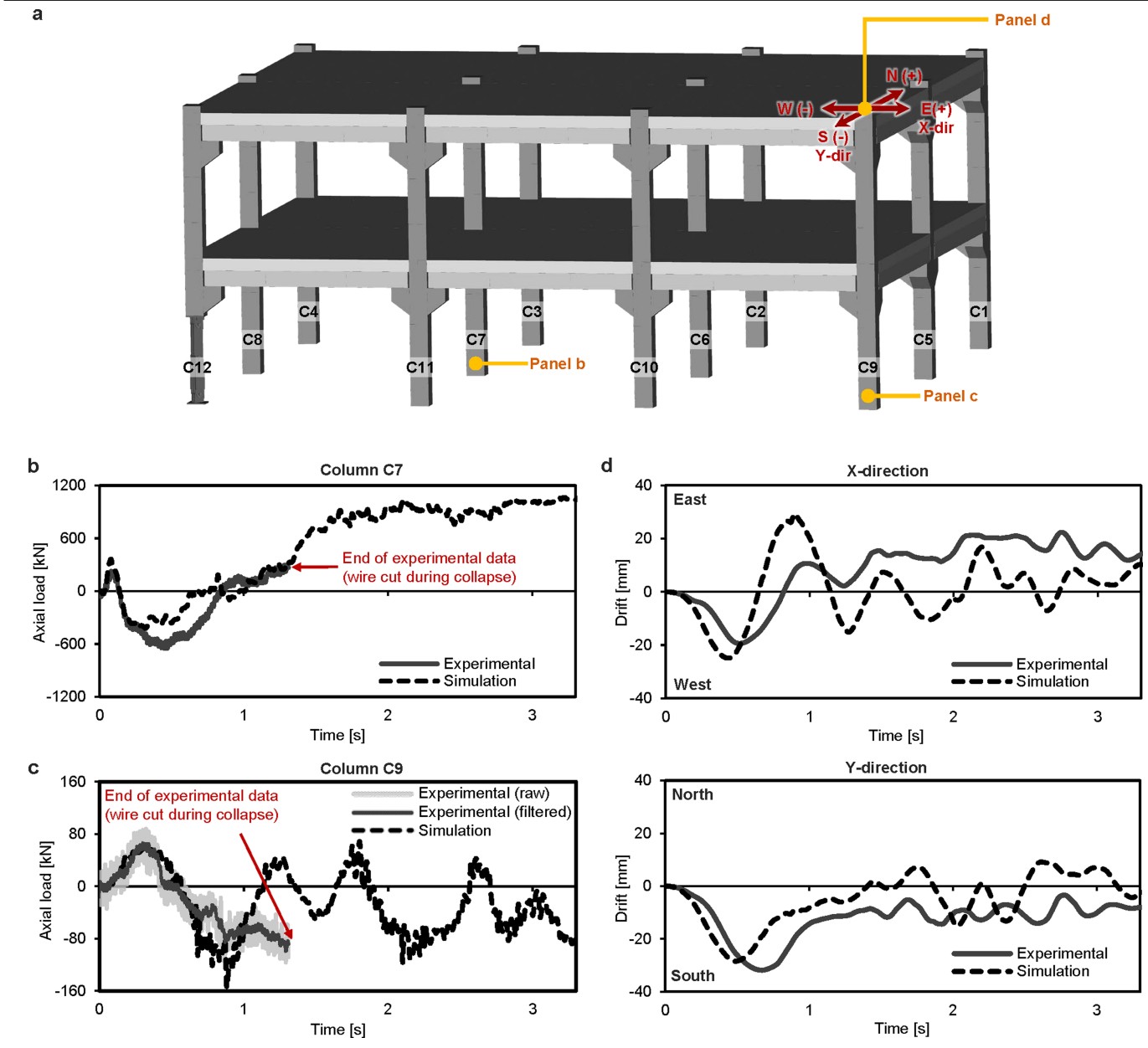

**Extended Data Fig. 2 | Comparison of measured experimental data and simulation predictions. a,** Location of shown comparisons. All data shown in panels **b** to **d** refer to the change in structural response following the sudden removal of column C12 (after having removed columns C8 and C11 in a previous phase). **b,** Change in axial load in lower part of column C7. **c,** Change in axial load in lower part of column C9. **d,** Change in drift measured in both directions parallel to each building side.

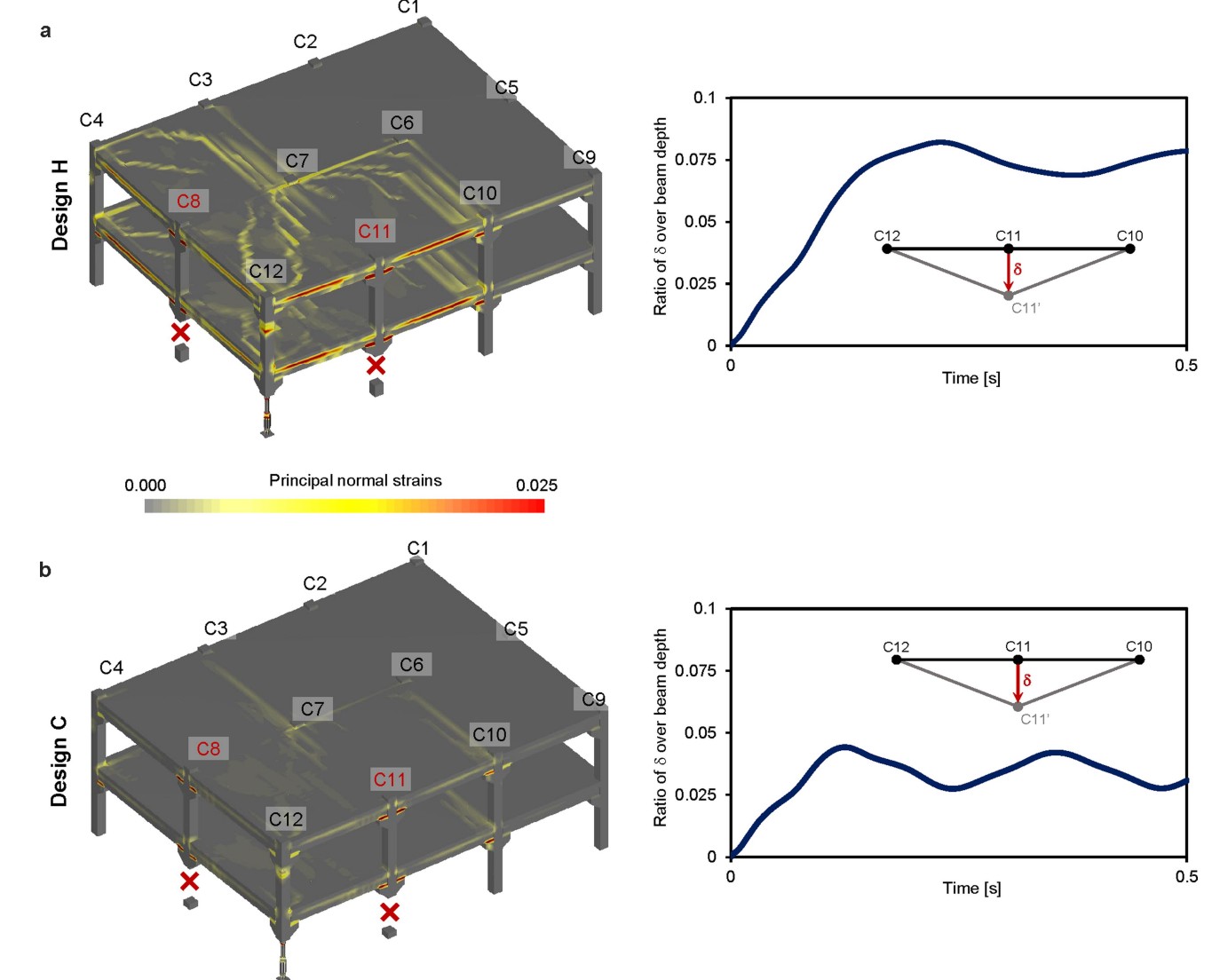

**Extended Data Fig. 3 | Computational simulations of Design H and Design C subjected to small initial failures.** Principal strains and relative vertical displacement at the location of column C11 after removal of columns C8 and C11 from Design H (**a**) and Design C (**b**).

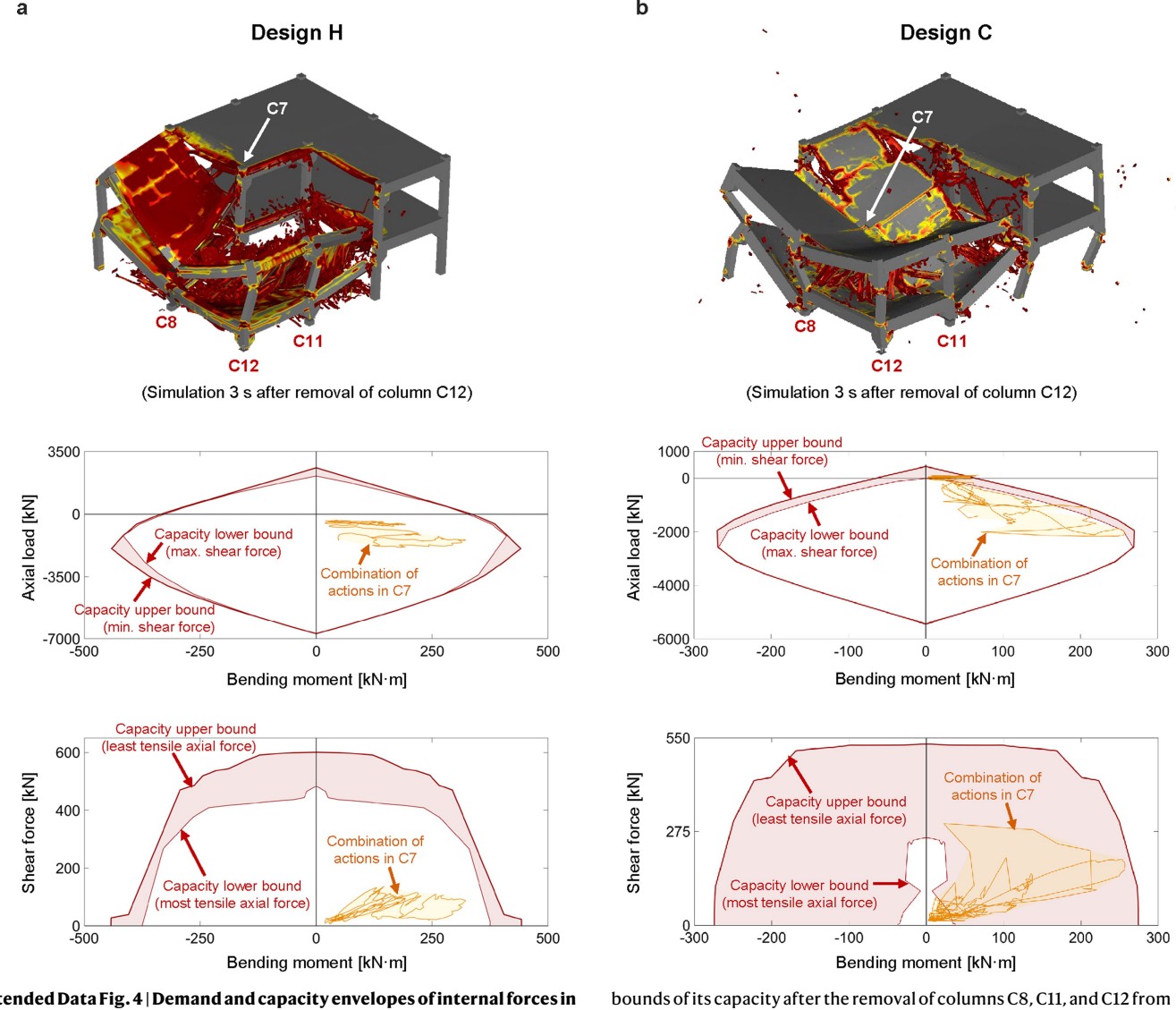

**Extended Data Fig. 4 | Demand and capacity envelopes of internal forces in Designs H and C subjected to large initial failures.** Evolution of axial loads, bending moments, and shear forces in column C7 compared to lower and upper bounds of its capacity after the removal of columns C8, C11, and C12 from Design H (**a**) and Design C (**b**).

## Scenario 1

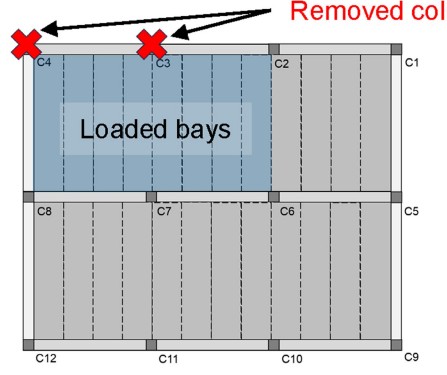

Removed columns

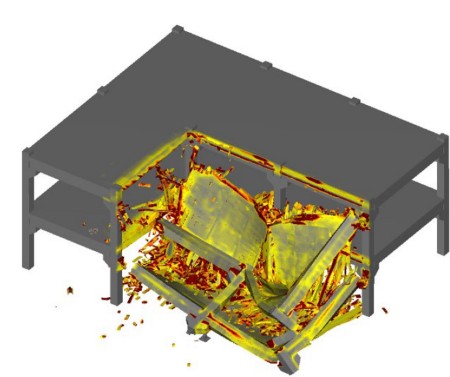

## Scenario 2

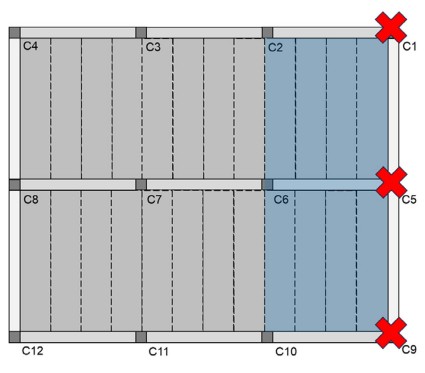

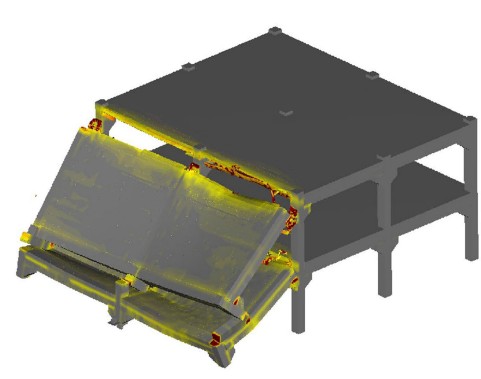

## Scenario 3

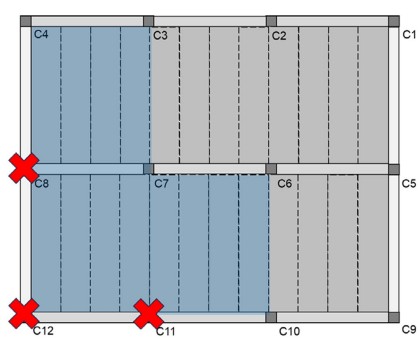

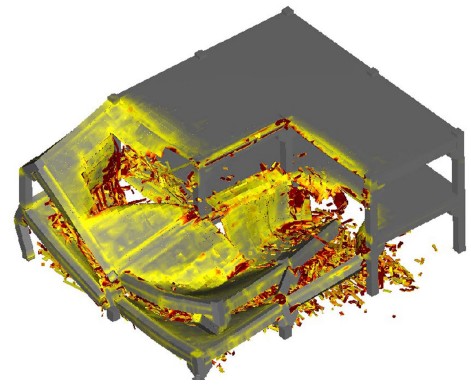

**Extended Data Fig. 5 | Initial failure scenarios considered for testing.** Simulation of three different initial failure scenarios that were considered for testing. Scenario 3 was selected for the experimental test.

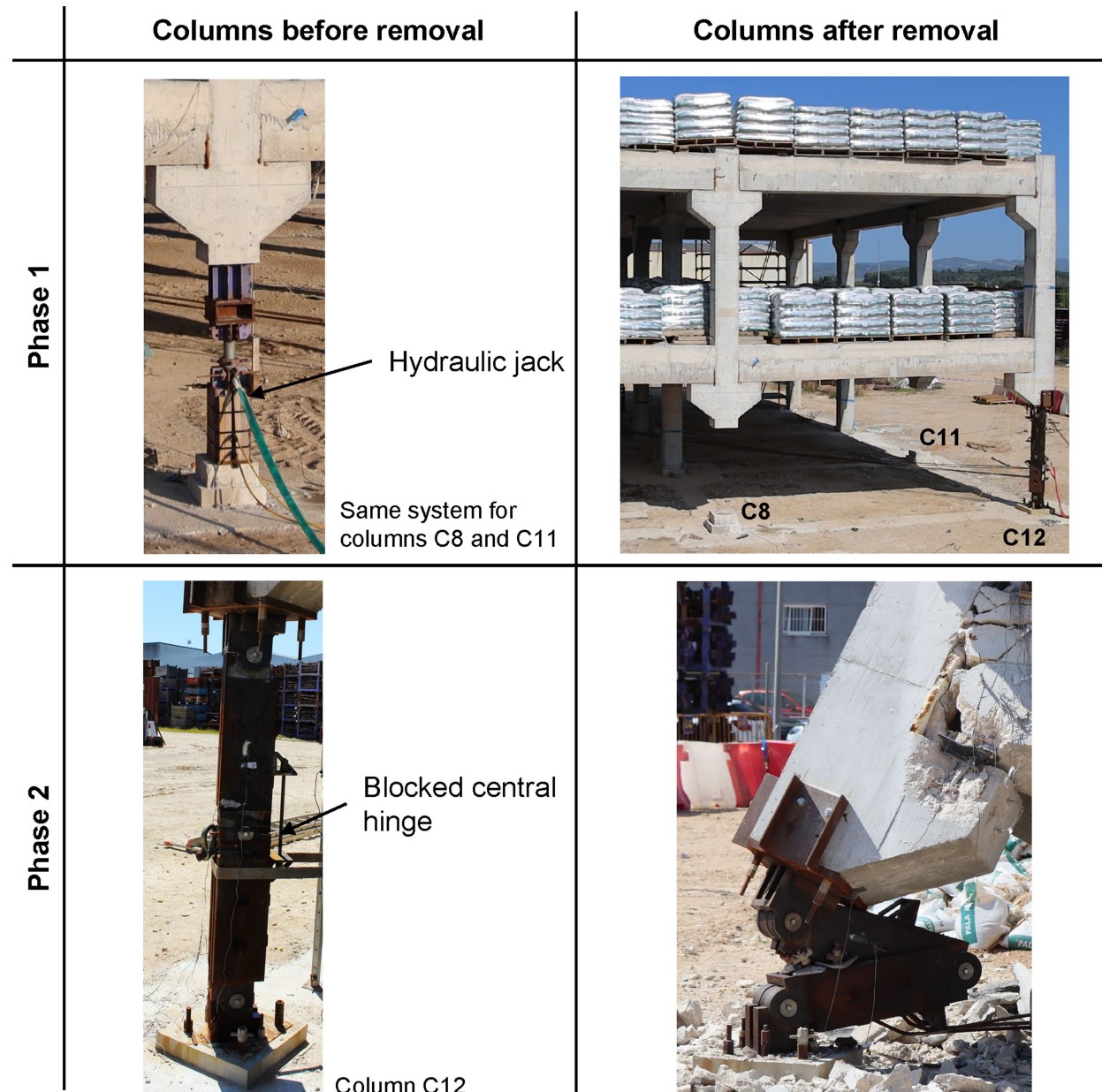

| Columns before removal | Columns after removal |

**Phase 1**

Hydraulic jack

Same system for columns C8 and C11

C11
C8
C12

**Phase 2**

Blocked central hinge

Column C12

**Extended Data Fig. 6 | Specially designed removable supports to perform column removals.** Removable supports designed for quasi-static column removals in phase 1 and sudden column removal in phase 2.

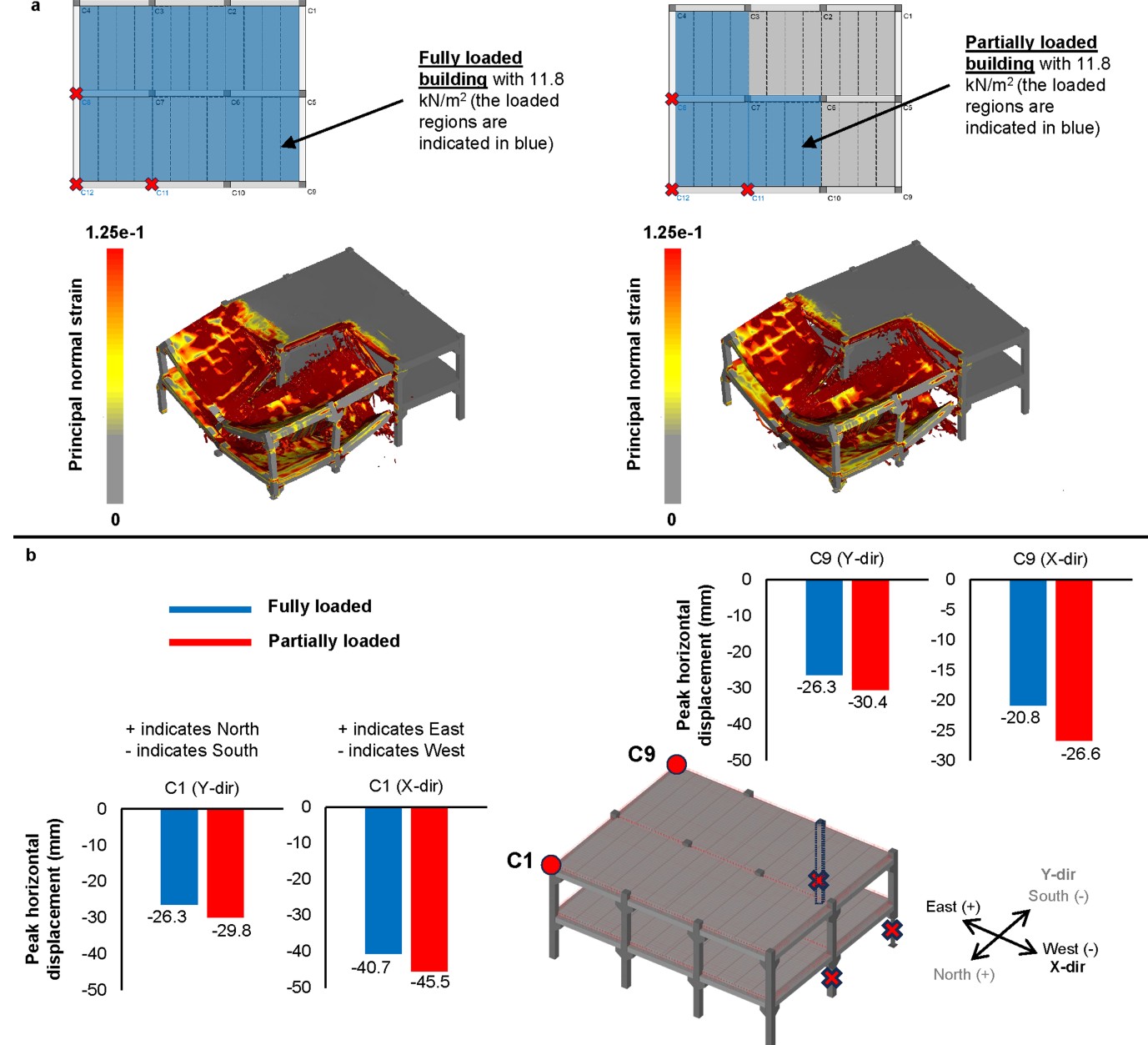

**Extended Data Fig. 7 | Comparison of simulations of fully loaded and partially loaded building specimen. a,** Loaded bays, deformed shape, and principal normal strains following the sudden removal of column C12 (after having removed columns C8 and C11 in a previous phase). **b,** Horizontal displacement in the east-west and north-south directions at the top of columns C1 and C9 (2nd floor).

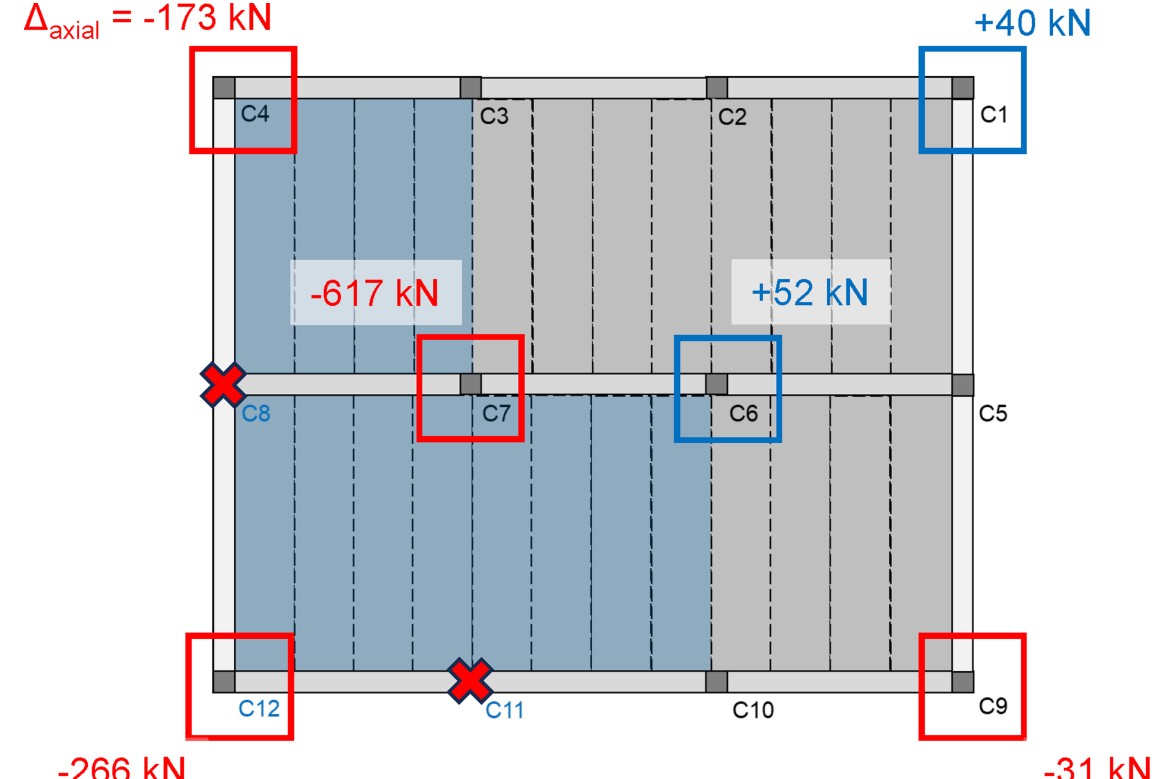

**Extended Data Fig. 8 | Measured redistribution of column axial forces during phase 1.** Maximum change in axial load of columns during phase 1 of testing based on recorded strain measurements.

**Extended Data Table 1 | Input parameters related to concrete materials for simulations based on the applied element method**

| Property | Value | Units | Notes |
|---|---|---|---|
| **Common properties for all concrete materials** | | | |
| Separation Strain | 0.10 | | |
| Internal friction coefficient | 0.80 | | |
| Specific Weight | 2.45E-06 | kg/mm$^3$ | |
| Poisson's ratio | 0.20 | | |
| External damping coefficient | 0.00 | | |
| Normal contact stiffness factor | 1.00E-04 | | |
| Shear contact stiffness factor | 1.00E-05 | | |
| Contact Spring unloading stiffness factor | 2.00 | | |
| Linear temperature expansion coefficient | 1.45E-05 | | |
| Failure softening factor | 0.10 | | |
| Shear stress failure weight | 1.00 | | |
| Residual shear strength factor | 1.00 | | |
| Tensile fracture energy | 0.0112 | kg/mm | Based on reported findings in https://doi.org/10.1007/s40069-014-0068-1 |
| Compressive fracture energy | 2.50 | kg/mm | Based on empirical relationships with compressive strength reported in https://doi.org/10.1016/j.cemconcomp.2018.06.015 |
| **Columns** | | | |
| Young's modulus | 3288 | kg/mm$^2$ | Based on material testing |
| Shear modulus | 1370 | kg/mm$^2$ | Isotropic behaviour assumed |
| Ultimate compressive strength | 3.39 | kg/mm$^2$ | Based on material testing |
| Ultimate tensile strength | 0.28 | kg/mm$^2$ | Based on material testing |
| Shear strength | 0.84 | kg/mm$^2$ | Taken as 3 times tensile strength |
| **Beams** | | | |
| Young's modulus | 3732 | kg/mm$^2$ | Based on material testing |
| Shear modulus | 1555 | kg/mm$^2$ | Isotropic behaviour assumed |
| Ultimate compressive strength | 4 | kg/mm$^2$ | Based on material testing |
| Ultimate tensile strength | 0.31 | kg/mm$^2$ | Based on material testing |
| Shear strength | 0.93 | kg/mm$^2$ | Taken as 3 times tensile strength |
| **Topping** | | | |
| Young's modulus | 3418 | kg/mm$^2$ | Based on material testing |
| Shear modulus | 1424 | kg/mm$^2$ | Isotropic behaviour assumed |
| Ultimate compressive strength | 3.48 | kg/mm$^2$ | Based on material testing |
| Ultimate tensile strength | 0.3 | kg/mm$^2$ | Based on material testing |
| Shear strength | 0.9 | kg/mm$^2$ | Taken as 3 times tensile strength |
| **Hollow-core planks (HA-45)** | | | |
| Young's modulus | 3670 | kg/mm$^2$ | Based on producer specifications |
| Shear modulus | 1529 | kg/mm$^2$ | Isotropic behaviour assumed |
| Ultimate compressive strength | 4.59 | kg/mm$^2$ | Based on producer specifications |
| Ultimate tensile strength | 0.39 | kg/mm$^2$ | Estimated using expression in Eurocode 2 |
| Shear strength | 1.17 | kg/mm$^2$ | Taken as 3 times tensile strength |

Assumptions adopted for estimating the input parameters are described. When no notes are provided, assumed values are based on information provided in the theoretical manual of the Extreme Loading for Structures software.

**Extended Data Table 2 | Input parameters related to steel materials for simulations based on the applied element method**

| Property | Value | Units | Notes |
|---|---|---|---|
| **B500 SD** | | | |
| Young's modulus | 20387 | kg/mm$^2$ | |
| Shear modulus | 7841 | kg/mm$^2$ | |
| Separation Strain | 1.00 | | |
| Internal friction coefficient | 0.80 | | |
| Specific Weight | 7.85E-06 | kg/mm$^3$ | |
| External damping coefficient | 0.00 | | |
| Normal contact stiffness factor | 1.00E-04 | | |
| Shear contact stiffness factor | 1.00E-05 | | |
| Contact Spring unloading stiffness factor | 2.00 | | |
| Linear temperature expansion coefficient | 1.30E-05 | | |
| Elastic strain limit | 0.00276 | | |
| Yield stress | 56.200 | kg/mm$^2$ | Based on material testing |
| Ultimate / yield stress ratio | 1.230 | | Based on material testing |
| Ultimate strain | 0.21 | | Based on material testing. True fracture strain considered given that post peak behaviour is governed by fracture energy. |
| Failure softening factor | 0.10 | | |
| Shear stress failure weight | 1.00 | | |
| Residual shear strength factor | 1.00 | | |
| Tensile / compressive fracture energy | 150 | kg/mm | Based on reported findings in https://doi.org/10.1023/A:1007500325074 |
| **Dowel bars** | | | |
| Shear modulus | 170 | kg/mm$^2$ | Reduced shear stiffness to account for cracking and allow localised deformations occurring at interface between corbels and beams. |
| *(all other properties are the same as B500 SD material)* | | | |
| **Y 1860 S7 (pretensioned strands)** | | | |
| Yield stress | 175.800 | kg/mm$^2$ | Based on producer specifications |
| Ultimate / yield stress ratio | 1.100 | | Based on producer specifications |
| Ultimate strain | 0.04 | | Based on producer specifications |
| *(all other properties are the same as B500 SD material)* | | | |

Assumptions adopted for estimating the input parameters are described. When no notes are provided, assumed values are based on information provided in the theoretical manual of the Extreme Loading for Structures software.