## [Peer Review File · Nature]

Manuscript Title: Arresting failure propagation in buildings through collapse isolation

Reviewer Comments & Author Rebuttals

Reviewer Reports on the Initial Version:

Reviewers' comments:

Referee #1 (Remarks to the Author):

A – The key results presented in the last section on “Discussion and future outlook”

B – The work is original and novel. No previous work on this scale has been conducted on building segmentation to the reviewers knowledge.

C - Overall, the manuscript is very well written and presents a novel and important approach to limiting building collapse propagation. The analysis approach and experimental testing is sound and well presented.

D – There is little use of statistics and uncertainties in the paper.

E – The conclusion are suitable for the test structure and building type tested only. Additional testing and analysis would be needed to verify the applicability of the hierarchical failure approach for other buildings.

F - There are a few suggestions to further improve the manuscript.

The first sentence in the introduction indicates that disasters have led to significant economic consequences and loss of life and many losses are attributed to building collapses. However, this is misleading. The hierarchical failure approach presented in this paper would not have limited the severity of most of these collapses as the collapses often occurred in poorly designed (unreinforced masonry) buildings under earthquakes. Additional clarification on the potential impact of the failure approach would be beneficial to help the reader accurately gauge the impact of the study.

The hierarchical failure approach is based on limiting the failure of columns, however in some structures, like flat plate buildings, the columns may not fail yet complete collapse does occur as in the Sampoong Department Store collapse in South Korea. While the failure approach is a novel and beneficial idea, as it is presented it is only relevant to some building types. Rather than alluding that the approach will work for all buildings, the authors should specify which types their current approach works for and then may indicate that other hierarchical paths could be developed for other building types. Specifically the authors should clarify that the current study and approach is for precast buildings.

Line 118 indicates that the partial loading was more critical as it produced greater drifts than then full loading case. However, in the case of partial loading there would be more unbalanced shear that could serve to help sever the structure at the loading edges. (Due to the dynamic movements of the load the momentum in the loaded portion may help to slice the structure at the edges.) The extended data only indicates the drifts. Was there any analysis to consider the overall failure effect?

Line 188 indicated that the building would have failed due to the unbalanced forces had the hierarchy -based isolation not been implemented. Is there analysis to support this statement?

The discussion looked at the comparison to the hierarchical approach and conventional current design. Was there any comparison to older designs before tie forces were implemented?

G – References are well cited.

H – Work is well presented and clear. The authors have done an excellent job of taking a

complicated engineering topic and making it more readable to a wider scientific community.

Referee #2 (Remarks to the Author):

A. Summary of the key results

The manuscript by Makoond et al. reports an interesting, say unique, experimental campaign to investigate the effectiveness of a new design philosophy towards progressive collapse. The authors show that a hierarchical-based design, which differs from the ordinary design suggested in the guidelines to enhance structural robustness, can improve the overall capacity of the structure to resist large initial damages. The experiment goes in this direction, with promising results.

B. Originality and significance: if not novel, please include reference

The study is novel and the experimental campaign is almost unique. The study can be the starting point for new design approaches in structural robustness studies, hence its significance is high.

C. Data & methodology: validity of approach, quality of data, quality of presentation

The quality of the manuscript is high, compared to the fact that it presents the results of a real scale collapse test. Such tests are rare, say unique in the robustness research panorama. Some points raise:

C1. It should be noted that the authors use their full-scale test to validate the results obtained through a AEM software. This is correct, showing that the reality and simulations are coherent and similar. Anyway, it should be mentioned that no considerations were made on the validity of the simulations related to the "Conventional" design method. It is shown that the Conventional method would lead to the complete collapse of the structure. This issue does not invalidate the quality of the obtained results, but the strength of the conclusions can be reduced. The Authors should add some considerations on the effective validity of the numerical analysis performed on the Conventional structure.

C2. The "quasi-static" removal of the Phase 1 must be justified. Such type of removal is unusual in robustness and progressive collapse studies since the dynamical effects are not considered. It may happen that a sudden removal in Phase 1 would lead to a completely different collapse type.

C3. The failure of the connections lies in the design of the dowel bars, hence in the shear capacity of such elements as they are the keys for transferring catenary forces between the beams.

Differentiating the capacity of beam-column connection would necessarily lead to compartmentalization of the components. Hence the system act as a fuse element. This point can be mentioned in the manuscript.

C4. What would have happened if all the structure, not half as the Authors did during the experiment, had been loaded? Has this point effects on the final configuration?

C5. The authors must specify why a corner column has been removed, rather than an inner column.

D. Appropriate use of statistics and treatment of uncertainties

The data have been correctly presented. It would be interesting to know if filtering was performed on the raw data.

E. Conclusions: robustness, validity, reliability

The conclusions are supported by the results. It is worth to define which are the "bounds of validity" of the proposed approach. Are there some cases in which the system does not work (bi-directional slabs)? Is the approach suitable when primary and secondary frames are well identified? What would happen if a middle column would have been removed? Is the system valid for corner columns, only?

F. Suggested improvements: experiments, data for possible revision

It is suggested to address the points reported in C.

G. References: appropriate credit to previous work?

Considering that the usual number of references to be included in a manuscript in Nature is limited, the Authors referenced the most relevant literature.

H. Clarity and context: lucidity of abstract/summary, appropriateness of abstract, introduction and conclusions

The text is clear and well organized. The main sections are clearly written and easy to follow. The use of the specific terms is limited to the essential, hence the manuscript is accessible to a large audience.

Author Rebuttals to Initial Comments:

Reviewer Comments & Author Response

REFEREE #1

A – The key results presented in the last section on “Discussion and future outlook”

B – The work is original and novel. No previous work on this scale has been conducted on building segmentation to the reviewers knowledge.

C - Overall, the manuscript is very well written and presents a novel and important approach to limiting building collapse propagation. The analysis approach and experimental testing is sound and well presented.

We thank the reviewer for these positive comments.

D – There is little use of statistics and uncertainties in the paper.

Indeed, the nature of the problem being addressed did not require extensive use of statistical methods to be able to draw reliable and meaningful conclusions. Specifically, modelling assumptions were validated by ensuring that the experimental test captured relevant collapse phenomena and one building was tested.

E – The conclusion are suitable for the test structure and building type tested only. Additional testing and analysis would be needed to verify the applicability of the hierarchical failure approach for other buildings.

We thank the reviewer for highlighting this very pertinent point. In fact, the fundamental conceptual basis of the proposed hierarchy-based collapse isolation approach is applicable to other structural types even if its physical implementation may vary. However, it was only applied to framed building structures as part of this work and more testing and analysis would be required to verify its applicability to other structural types. We have made a small modification to the conclusion (line 211) to clarify this point:

“This paves the way for the proposed solution to be developed, tested, and implemented in different building types with different building elements.”

We have also included the following sentence in the introduction (line 34) to clarify the scope and avoid any misinterpretation:

“In this work, hierarchy-based collapse isolation is applied to framed building structures.”

F - There are a few suggestions to further improve the manuscript.

The first sentence in the introduction indicates that disasters have led to significant economic consequences and loss of life and many losses are attributed to building collapses. However, this is misleading. The hierarchical failure approach presented in this paper would not have limited the severity of most of these collapses as the collapses often occurred in poorly designed (unreinforced masonry) buildings under earthquakes. Additional clarification on the potential impact of the failure approach would be beneficial to help the reader accurately gauge the impact of the study.

We completely agree with the reviewer that the proposed approach can require a different physical implementation for different building types. The purpose of the first sentence is to highlight the tremendous cost of extreme events and why we need to develop solutions to improve building resilience. We hope that the modifications made to the revised manuscript to clarify the scope of the work (line 34) and future research needs (line 211) will prevent any misconception.

The hierarchical failure approach is based on limiting the failure of columns, however in some structures, like flat plate buildings, the columns may not fail yet complete collapse does occur as in the Sampoong Department Store collapse in South Korea. While the failure approach is a novel and beneficial idea, as it is presented it is only relevant to some building types. Rather than alluding that the approach will work for all buildings, the authors should specify which types their current approach works for and then may indicate that other hierarchical paths could be developed for other building types. Specifically the authors should clarify that the current study and approach is for precast buildings.

As previously mentioned, we agree with the reviewer that applying the proposed approach to other building types including flat slabs would require more analysis and testing. To this end, we hope the modifications made in the introduction (line 34) and conclusion (line 211) will clarify the aims and scope of the present work.

Line 118 indicates that the partial loading was more critical as it produced greater drifts than then full loading case. However, in the case of partial loading there would be more unbalanced shear that could serve to help sever the structure at the loading edges. (Due to the dynamic movements of the load the momentum in the loaded portion may help to slice the structure at the edges.) The extended data only indicates the drifts. Was there any analysis to consider the overall failure effect?

In fact, our analyses revealed that both the final collapse state (shown in Extended Data Fig. 7) and the failure mode at the segment border would be almost identical when comparing the partially loaded and fully loaded scenarios. This is exemplified by examining the evolution of stresses in key reinforcing bars around column C7, from which the separation initiated (see Fig. R1 below). It is clear to see that differences at the segment border as the structure is severed are rather negligible when comparing these two loading scenarios. Since the differences in building drift were more significant, we considered that it would be more meaningful to test the stability of the remaining upright part of the building under more extreme conditions.

Several modifications have been implemented to explain this reasoning more clearly in this revision:

- A new supplementary video (Supplementary Video 3) has been provided comparing simulations of the partially loaded and fully loaded structure.
- The text in the Methods section from lines 237 to 245 has been modified:
“We performed additional computational simulations to compare this partial loading configuration and loading the entire building. The simulations indicated that both would have resulted in almost identical final collapse states (Extended data Fig. 7). However, the partial loading configuration introduced a higher magnitude of unbalanced moment to surrounding columns, which induces more critical bending and shear in columns. Simulations confirmed that the lateral drift of the remaining part of the building would be higher when only three bays are loaded, indicating that its stability would be tested to a greater extent with this loading configuration (Extended data Fig. 7).”

- The following explanation has been included in the main text from lines 122 to 127:
“Predictive simulations indicated that the failure mode and overall collapse would be almost identical when comparing this partial loading configuration to one whereby the entire building is loaded (Supplementary Video 3). However, the partial loading configuration can be considered as being more critical for the part of the structure expected to remain upright due to the greater drifts it produces during collapse (Methods, ‘Experiment and monitoring design’; Extended data Fig. 7).”

Fig. R1 | Comparison of simulations of fully loaded and partially loaded building specimen. a. Loaded bays, deformed shape, and principal normal strain following the sudden removal of column C12 (after having removed columns C8 and C11 in a previous phase). **b.** Comparison between two loading scenarios of maximum axial stress in continuity bars and shear stress in dowel bars of column C7 on the side of the collapse.

Line 188 indicated that the building would have failed due to the unbalanced forces had the hierarchy-based isolation not been implemented. Is there analysis to support this statement?

Yes, the analysis comparing the performance of the hierarchy-based collapse isolation design to that of a conventional design supports this statement. It is presented in the “Hierarchy-based collapse isolation” section of the main text, specifically from lines 91 to 96 of the revised manuscript and is further supported by Extended data Fig. 4 and Supplementary Video 1.

The discussion looked at the comparison to the hierarchical approach and conventional current design. Was there any comparison to older designs before tie forces were implemented?

If no tying systems are implemented, collapse can occur even after small initial failures, and the structural design of the building would not comply with minimum robustness requirements in current building codes. Since the proposed approach is intended for new designs that must comply with current robustness requirements, we did not perform any comparison with older designs before tie forces were implemented.

G – References are well cited.

H – Work is well presented and clear. The authors have done an excellent job of taking a complicated engineering topic and making it more readable to a wider scientific community.

We are grateful for the reviewer's positive opinion of our work.

REFEREE #2

A. Summary of the key results

The manuscript by Makoond et al. reports an interesting, say unique, experimental campaign to investigate the effectiveness of a new design philosophy towards progressive collapse. The authors show that a hierarchical-based design, which differs from the ordinary design suggested in the guidelines to enhance structural robustness, can improve the overall capacity of the structure to resist large initial damages. The experiment goes in this direction, with promising results.

B. Originality and significance: if not novel, please include reference

The study is novel and the experimental campaign is almost unique. The study can be the starting point for new design approaches in structural robustness studies, hence its significance is high.

We thank Referee #2 for these positive comments.

C. Data & methodology: validity of approach, quality of data, quality of presentation

The quality of the manuscript is high, compared to the fact that it presents the results of a real scale collapse test. Such tests are rare, say unique in the robustness research panorama. Some points raise:

C1. It should be noted that the authors use their full-scale test to validate the results obtained through a AEM software. This is correct, showing that the reality and simulations are coherent and similar. Anyway, it should be mentioned that no considerations were made on the validity of the simulations related to the "Conventional" design method. It is shown that the Conventional method would lead to the complete collapse of the structure. This issue does not invalidate the quality of the obtained results, but the strength of the conclusions can be reduced. The Authors should add some considerations on the effective validity of the numerical analysis performed on the Conventional structure.

It is worth mentioning that comparisons between experimental and simulation results presented in scientific literature¹⁻³ clearly demonstrate the ability of the Applied Element Method (AEM) to accurately represent collapse phenomena if suitable modelling assumptions are made. In this case, the good agreement obtained with the test results show that collapse simulations performed with the AEM can be considered as being reliable for the type of building studied when adopting the rational basis summarised in Methods (Modelling strategy) and Extended Data Tables 1 and 2. It is true that for structures made of substantially different materials and characterised by significantly different dynamic loading, different modelling assumptions might have to be made to ensure a valid representation. However, there are no such differences between the conventional design simulated in this work and the tested building. As such, the reported predictions can be considered as being sufficiently reliable.

As suggested by the reviewer, we have made the following modifications to clarify the effective validity of the numerical analyses shown in this work:

- The sentence starting on line 76 of the main text has been modified:
*"The ability of these simulations to accurately represent collapse phenomena **for the type of building being studied** was later validated by comparing its predictions to the structural response observed during a purposely designed collapse test of a full-scale building (Extended data Fig. 2; Supplementary Video 6)."*

- A comment has been added from lines 200 to 203 of Methods section:
“To validate these assumptions for simulating the precast building designs described previously, it was ensured that the full-scale test performed as part of this work captured all relevant phenomena influencing collapse (large displacements, fracture, contact, and collision).”

C2. The “quasi-static” removal of the Phase 1 must be justified. Such type of removal is unusual in robustness and progressive collapse studies since the dynamical effects are not considered. It may happen that a sudden removal in Phase 1 would lead to a completely different collapse type.

We thank the Reviewer for raising this highly relevant point. In fact, we accounted for possible dynamic effects during Phase 1 of testing by deliberately increasing the imposed gravity loads by almost twice that specified in Eurocodes⁴ for accidental design situations (line 120 of main text).

The aim of Phase 1 was to test the ability of the structure to develop stable alternative load paths and prevent collapse initiation after small initial failures. Building codes specify the imposed loads to be used when analysing such scenarios. If a structure can find a state of equilibrium after the removal of a column, the expected dynamic displacement can be accurately estimated by performing a static analysis but imposing a load that is multiplied by a dynamic amplification factor (DAF)⁵. The upper bound of possible DAFs is 2, and it is typically more appropriate to use lower values for most structures^{6,7}. Therefore, since a DAF of 2 was used in Phase 1, the outcome clearly shows that the building design complies with robustness requirements in current building standards (lines 132 to 133 of the main text). Therefore, the loading protocol used in Phase 1 and the corresponding results are still valid and meaningful.

C3. The failure of the connections lies in the design of the dowel bars, hence in the shear capacity of such elements as they are the keys for transferring catenary forces between the beams. Differentiating the capacity of beam-column connection would necessarily lead to compartmentalization of the components. Hence the system act as a fuse element. This point can be mentioned in the manuscript.

Lines 49 to 51 of the revised Methods section has been modified as follows based on the suggestion of the reviewer:

“The design of the dowel bars is one of the key aspects in achieving partial-strength connections that fail at a specific threshold to enable segmentation.”

C4. What would have happened if all the structure, not half as the Authors did during the experiment, had been loaded? Has this point effects on the final configuration?

The expected failure mode and final collapse state are expected to be almost identical when comparing the fully loaded and partially loaded test structure. A new Supplementary video (Supplementary Video 3) has been provided and minor modifications have been made in the main text (lines 122 to 127) and Methods section (lines 237 to 240) to clarify this fact in the submitted revision:

- Lines 122 to 125 of the main text:
“Predictive simulations indicated that the failure mode and overall collapse would be almost identical when comparing this partial loading configuration to one whereby the entire building is loaded (Supplementary Video 3).”
- Line 237 to 240 of the Methods section:
“We performed additional computational simulations to compare this partial loading configuration and loading the entire building. The simulations indicated that both would have resulted in almost identical final collapse states (Extended data Fig. 7; Supplementary Video 3).”

C5. The authors must specify why a corner column has been removed, rather than an inner column.

It is important to highlight that the removal of the corner column C12 was performed after the two penultimate columns surrounding it were removed. This does not represent only a corner column failure but rather an initial failure which is large enough to trigger collapse initiation (see main text lines 41 to 45). Edge and corner regions of the building were prioritised for this study since they are usually exposed to a wider range of external threats⁸⁻¹¹. This has now been mentioned from lines 210 to 212 of the revised Methods section. The specific initial failure scenario tested was chosen after evaluating and comparing several possible initial failure scenarios in edge and corner regions of the test building (Extended data Fig. 5; Methods, 'Experiment and monitoring design'; Supplementary Video 2).

D. Appropriate use of statistics and treatment of uncertainties

The data have been correctly presented. It would be interesting to know if filtering was performed on the raw data.

We used a conventional low-pass filter at 40 Hz for electrical sensors and no filtering for fibre-optic sensors.

E. Conclusions: robustness, validity, reliability

The conclusions are supported by the results. It is worth to define which are the "bounds of validity" of the proposed approach. Are there some cases in which the system does not work (bi-directional slabs)? Is the approach suitable when primary and secondary frames are well identified? What would happen if a middle column would have been removed? Is the system valid for corner columns, only?

Yes, the general philosophy of hierarchy-based collapse isolation can be applied to other types of structural systems. Naturally, its implementation could differ significantly depending on the type of structure. In this work, it has been applied to framed building structures. This has now been clarified in line 34 of the introduction, while the need for more testing in order to be able to apply the approach to other systems is highlighted in line 211 of the revised main text.

The proposed approach is also completely applicable even if the initial failure happens in other parts of the building (not only in corner regions). It is a threat-independent approach for arresting collapse propagation irrespective of where the initial failure initiates. However, we deliberately selected a large initial failure in the exterior region of the building for our test campaign after evaluating several possible options (described from lines 210 to 229 in the Methods section).

F. Suggested improvements: experiments, data for possible revision

It is suggested to address the points reported in C.

All suggested improvements reported in C have been addressed and modifications have been made to the main text, the Methods Section, and to the Supplementary material accordingly.

G. References: appropriate credit to previous work?

Considering that the usual number of references to be included in a manuscript in Nature is limited, the Authors referenced the most relevant literature.

H. Clarity and context: lucidity of abstract/summary, appropriateness of abstract, introduction and conclusions

The text is clear and well organized. The main sections are clearly written and easy to follow. The use of the specific terms is limited to the essential, hence the manuscript is accessible to a large audience.

We are grateful for the reviewer's positive opinion on the clarity of our work.

References employed in our response to referees

1. Tagel-Din, H. & Meguro, K. Nonlinear simulation of RC structures using applied element method. *Doboku Gakkai Ronbunshu* **2000**, 13–24 (2000).
2. Grunwald, C. *et al.* Reliability of collapse simulation – Comparing finite and applied element method at different levels. *Eng Struct* **176**, 265–278 (2018).
3. Applied Science International. Extreme Loading for Structures - Theoretical Manual. 108 <https://www.extremeloading.com/> (2021).
4. European Committee for Standardization (CEN). EN 1990:2002: Eurocode 0 - Basis of structural design. Preprint at (2002).
5. Izzuddin, B. A., Vlassis, A. G., Elghazouli, A. Y. & Nethercot, D. A. Progressive collapse of multi-storey buildings due to sudden column loss — Part I: Simplified assessment framework. *Eng Struct* **30**, 1308–1318 (2008).
6. Marchand, K., McKay, A. & Stevens, D. J. Development and Application of Linear and Non-Linear Static Approaches in UFC 4-023-03. in *Structures Congress 2009* 1–10 (American Society of Civil Engineers, Reston, VA, 2009). doi:10.1061/41031(341)191.
7. Department of Defense (DoD). *UFC 4-023-03. Design of Buildings To Resist Progressive Collapse. Design of Buildings To Resist Progressive Collapse* 34–37 (2016).
8. Makoond, N., Shahnazi, G., Buitrago, M. & Adam, J. M. Corner-column failure scenarios in building structures: Current knowledge and future prospects. *Structures* **49**, 958–982 (2023).
9. Adam, J. M., Buitrago, M., Bertolesi, E., Sagaseta, J. & Moragues, J. J. Dynamic performance of a real-scale reinforced concrete building test under a corner-column failure scenario. *Eng Struct* **210**, 110414 (2020).
10. Starossek, U. *Progressive Collapse of Structures. Progressive Collapse of Structures* (ICE Publishing, 2017). doi:10.1680/pcos.61682.
11. Zhao, Z., Guan, H., Li, Y., Xue, H. & Gilbert, B. P. Collapse-resistant mechanisms induced by various perimeter column damage scenarios in RC flat plate structures. *Structures* **59**, 105716 (2024).

Reviewer Reports on the First Revision:

Referees' comments:

Referee #1 (Remarks to the Author):

The authors have adequately addressed the points raised in the previous reviews. This reviewer has no more comments for the authors.

Referee #2 (Remarks to the Author):

The authors addressed the points raised during the first review round in an appropriate, sound and pertinent way. The reviewer has no more issues on the manuscript.

Author Rebuttals to First Revision:

Reviewer Comments & Author Response

REFEREE #1

The authors have adequately addressed the points raised in the previous reviews. This reviewer has no more comments for the authors.

We would like to thank the referee for his/her detailed review of the manuscript and for the comments provided which have contributed to enhancing the quality of the manuscript.

REFEREE #2

The authors addressed the points raised during the first review round in an appropriate, sound and pertinent way. The reviewer has no more issues on the manuscript.

We are grateful for the feedback provided by Referee #2, which helped us to improve the clarity of the article.